# Alternative Clinical Practice Experiences of Korean Nursing Students during the COVID-19 Pandemic

**DOI:** 10.3390/healthcare11172452

**Published:** 2023-09-01

**Authors:** Eun-Ho Ha

**Affiliations:** Department of Nursing, Jungwon University, Goesan-gun 28024, Republic of Korea; rnhaeunho@jwu.ac.kr

**Keywords:** clinical practicum, COVID-19, experience, learning, nursing student

## Abstract

**Background:** Routine clinical practice (RCP) experiences provide nursing students with the opportunity to gain confidence in their professional roles. The purpose of this study was to explore the alternative clinical practice (ACP) experiences of nursing students during the COVID-19 pandemic using the Q methodology. **Methods:** Fifty-two nursing students located in four cities across South Korea participated. The participants sorted the order of and ranked 41 Q statements about their experiences with ACP into a Q sort table; the data were analyzed using the PQ method. **Results:** The following three distinct viewpoints were extracted: (1) ACP helps with balancing learning and life (favorable view of ACP); (2) ACP does not help with balancing theory and clinical field practice (critical view of ACP); and (3) RCP cannot be replaced by ACP (negative view of ACP). **Conclusions:** The findings of this study indicate that developing a curriculum for clinical practice that can enhance the strengths of ACP while compensating for its weaknesses will help promote learning among nursing students.

## 1. Introduction

The World Health Organization (WHO) declared the widespread coronavirus disease of 2019 (COVID-19) as a pandemic on 11 March 2020 because it was a public health emergency of international concern [1,2]. Since then, COVID-19 has had a considerable impact on not only the lives of people globally but also politics, economy, culture, health, and education [3]. One of the most notable effects was in terms of education wherein face-to-face classes were converted to non-face-to-face classes as part of distance education efforts to prevent mass infection during the pandemic and to ensure social distancing [4,5]. In particular, online alternative clinical practice (ACP) was conducted for nursing students, which was unprecedentedly abrupt and without preparation; this has aroused social concerns and issues regarding the nurturing of future medical professionals.

Routine clinical practice (RCP) in nursing education is a traditional learning method that integrates nursing theory and practice [6]; it is an overarching learning opportunity for students to directly practice the diverse nursing skills acquired through classes [7,8]. RCP experiences provide nursing students with the opportunity to gain confidence in their professional roles and become competent nurses in the future [8,9].

ACP was designed as an online learning method to replace RCP during the COVID-19 period and typically uses platforms such as virtual simulations [10], miscellaneous virtual scenarios with simulations [11], telehealth clinical practice [6], online game-based learning [12], online nursing practicum developed by instructors [13], and a nursing skills website developed by a certain company. The greatest advantage of the ACP learning methods is that online network environments provide convenient and flexible access to nursing students anywhere and anytime without concerns regarding the COVID-19 infection [4]. Despite these benefits and the application of various learning methods using online platforms, discrepancies were found between the learning objectives and learning achievement in ACP [14]. Specifically, a decline in nursing professionalism and clinical adaptability, practical nursing skills, and quality of patient care are the major concerns in ACP [15,16]. Nursing students’ psychological issues, such as stress and fear stemming from RCP, should also not be ignored [8,17,18,19]. The inappropriate execution of ACP can negatively affect professors’ online educational methodologies and students’ professional identities in addition to clinical competency in the short or long term [20]. Appropriate execution of ACP, however, provides students the opportunity for improved self-directed learning [16] and professors the chance to develop various online nursing virtual platforms, such as Skillsnasium created by the Mount Carmel College of Nursing in the United States [21].

Many qualitative and quantitative studies have been conducted worldwide to explore ACP among nursing students during the COVID-19 pandemic [12,22,23]. These studies have highlighted that the clinical education of nursing students should be conducted systematically without omissions under any circumstances because nursing students are the frontline leaders responsible for healthcare systems in the future. Previous studies have also emphasized the need for continuous development of diverse ACP educational platforms in preparation for future pandemic situations [8,15,18,21]. Thus, it is vital to discuss the present and future preparations for ACP learning methods for nursing students beyond educational and cultural differences. However, the most important consideration is to improve the current ACP and develop a new, robust ACP method based on the experiences of nursing students. The experiences and attitudes of nursing students toward ACP are key factors that can positively or negatively affect learning outcomes.

There has been a notable rise in the Q methodology in nursing education literature and research over the past few years. Q methodology is a research method that is optimized to understand the subjective experiences and attitudes of nursing students toward ACP in a qualitative manner and to analyze them objectively and quantitatively. Q studies online in the era of COVID-19 have the advantage of being able to recruit more participants from different regions and provide a COVID-19 secure environment by eliminating face-to-face surveys [24]. The aim of the present study was to explore the experiences of South Korean nursing students who participated in ACP during the COVID-19 pandemic using the Q methodology.

## 2. Methods

### 2.1. Overview of Q Methodology (Q)

The Q methodology was first introduced in 1935 by the psychologist William Stephenson. It is a research method designed to systematically investigate individual subjectivity (thoughts, attitudes, points of view, perspectives, ideas, etc.) on various issues involving humans [24]. In particular, Q combines qualitative and quantitative methods, and it also has the substantial advantage of being able to quantitatively categorize human internal perspectives [25]. Because of this, Q is widely used as a research method in many areas, including medical service, business, education, advertisement, politics, environment, and economy [26]. However, there is a possible limitation that cannot be ruled out that the steps involved in the Q study can be time-consuming [24].

### 2.2. Research Procedure

Figure 1 displays the five steps of the Q research procedure. The inclusion criteria for the nursing students who participated in this study were as follows: (1) third- and fourth-year students enrolled in a College of Nursing in six regions in South Korea; (2) students who had experience with ACP.

#### 2.2.1. Step 1: Concourse (Process of Creating a Q Set)

The first step in Q research is to create a concourse called the Q set. The concourse includes everything that can be expressed as a topic and is theoretically inexhaustible [25]. Akhtar-Danesh et al. [26] reported that concourses could be obtained from various sources. In this study, expert opinions and arguments, previous studies, newspapers, and news about remote learning during COVID-19 were reviewed first to develop the Q set. Fifty nursing students voluntarily applied for the focus group interview through the J University website, and these participants were then divided into eight groups (six to seven students per group). The participants can freely express their experiences, thoughts, and attitudes candidly and in response to the semi-structured questions about ACP prepared by the researcher (Ph.D.) in advance. The semi-structured interview questions were as follows:What was the most difficult thing you experienced during ACP in the COVID-19 pandemic?What was the greatest advantage you experienced during ACP?What suggestions would you provide if RCP were replaced by ACP?What efforts should universities and nursing department professors and staff make for efficient ACP?What efforts should professors and students make for ACP to be effective?

The interviews continued until a saturation point was reached (about an hour or two). All interviews were recorded with the nursing students’ permission, and all recordings were transcribed by research assistants (graduate students). A total of 118 statements were primarily extracted in the first step. Ambiguous or redundant statements were removed or revised by a group of four experts consisting of one Q methodologist, two nursing professors, and one nurse supervisor, such that the 118 initial statements were finally reduced to 68.

#### 2.2.2. Step 2: Q Set (Process of Developing a Final List of Statements)

The second step in Q research is to further condense the statements obtained in the first step to finalize a list of statements called the Q set. Each statement in the Q set is refined to be more concise and understandable through this process. The number of statements in the Q set may vary. However, this can be controversial because the larger the number of statements, the more is the time required. Akhtar-Danesh et al. [26] reported that a study with 50 statements took about 30 to 60 min to sort. Brown [27] suggested that 40 to 50 statements would be adequate for a Q set. Based on this, the panel of four experts and the research team reviewed the 68 statements to reidentify similarities and redundancies in the sentences and confirmed 41 statements as the final Q set (Table 1). Before the next step (Q sort), a pilot study was conducted to verify the performance of the Q set using six volunteer nursing students (three each in the third and fourth years).

#### 2.2.3. Step 3: P Set (Process of Recruiting Study Participants)

The third step in Q research is to recruit participants to practically sort the Q set in the Q sort table. Q research generally prefers purposive sampling over random because it allows subjective perspectives on a certain topic or issue to be articulated. Furthermore, the Q method typically requires smaller sample sizes of around 40 to 60 compared to the larger sizes needed in quantitative studies. This is because Q research is not aimed at generalizing the study results [26]. In the present study, a convenient sample of 52 nursing students attending universities in six regions across South Korea who had ACP experience were invited to complete the Q sorting (face to face). All data (including 50 students in the Q-set) were collected from 10 March 2022 to June 2022.

#### 2.2.4. Step 4: Q Sort (Process of Ranking a Q Set)

The fourth step in Q research is to sort the order of and rank the Q set written on paper cards by the level of importance using a predefined grid called a Q sort table. Churruca et al. [25] reported that the numerical rankings should range from −4 to +4 (most agreeable, +4; neutral, 0; most disagreeable, −4). Before sorting and ranking the Q set, the research team explained the meaning of each statement card and the method of Q set distribution in the Q sort table grid to help the participants understand the process. Table 2 shows an example of a completed Q sort in this study. After sorting and ranking, the participants were asked to elaborate on why the statements were placed in the +4 or −4 squares. These post-sorting narratives could be relevant to comprehending the unique characteristics of each factor.

#### 2.2.5. Step 5: Factor Extraction and Interpretation (Process of Analyzing Collected Data)

The fifth step in Q research is to analyze the scores converted from the −4 to +4 scale to 1 to 9 using the PQ method. Q method uses software packages Version 2.35 such as pc QUANL, PQ method, Ken Q, or PCQ to analyze the data and extract factors.

The PQ method is one of the statistical programs optimized for the Q method, which offers various features of the extracted factors that represent groups of participants with similar viewpoints or perspectives. In the Q method, the percentage of the total variance, eigenvalues (greater than 1.0), and z-scores (greater than +1.0, positive or less than −1.0, negative) are important values relevant to the Q sort analysis. A distinguishing statement for a factor is one whose scores are significantly different (*p* < 0.01) from the scores of the other factors [26,28]. Such distinguishing statements can help identify the characteristics of each factor via subtle differences in the viewpoint of each factor [28]. Consensus statements can be useful for highlighting similarities between the factors.

### 2.3. Ethical Considerations

Ethical approval (1044297-HR-202109-006-02) was obtained from the Institutional Review Board (IRB) of J University before the study. The anonymity and confidentiality of participation were explained to the study participants. The participants were also informed that they could withdraw at any time if they wanted and that the collected data would be sealed and stored in the archives of the laboratory in charge of the study and not used for any purpose other than the research study. We obtained informed consent from the participants after explaining the purpose of the study as well as their voluntary participation.

### 2.4. Rigor

For the trustworthiness of the Q method, which combines qualitative and quantitative approaches, the corresponding research procedures and analysis were thoroughly followed [27]. A focus group interview was conducted using a semi-structured questionnaire to understand the subjectivity in the ACP experiences of nursing students for rigor of the qualitative aspects [29]. The validity of the Q method is verified through a process involving three stages: content validity, face validity, and Q-sorting validity. The reliability of the Q methodology is verified when there is a Pearson correlation coefficient of 0.80 or higher through test–retest [26,27]. To secure validity, the content and face validity from four experts and Q sorting by six nursing students were obtained (pilot test). To secure reliability, a test–retest procedure (by ten nursing students) was used at two-week intervals. The correlation coefficients between respective Q-sorts exceeded 0.83. To classify the subjectivity of the 52 nursing students, the PQ method was used for rigor regarding the quantitative aspects [26,27].

## 3. Results

Three distinct factors were extracted from the statements that explained the experiences of nursing students with ACP during the COVID-19 pandemic (Figure 1). Table 1 shows the grid positions of the 41 statements for these three factors. The asterisks (*) present distinguishing statements that are statistically significantly different (*p* < 0.01) from the scores of other factors. The bold statements show common views with positive or negative agreement for the three factors.

### 3.1. Factor I: ACP Helps with Balancing Learning and Life (Favorable View of ACP)

Eighteen (34.6%) out of the 52 participants were classified into the factor I group. Of these 18 participants, 13 (72.2%) were female, and six (33.3%) were under the age of 21 (Table 3).

Compared to the factor II and III groups, the factor I group agreed most (positive perspective) with statements 5 (feel more comfortable and at ease) and 28 (no risk of infection with COVID-19) with a score of +4. Factor I group disagreed most (negative perspective) with statements 27 (do not want to do the ACP again) and 30 (RCP should be an exception) with scores of −4 (Table 1). Table 4 shows examples of the post-sorting narratives on factor I, explaining why the participants sorted the statements from +4 to −4 on the grid. From this perspective, the responses to factor I reflected the view that ACP could be highly conducive to balancing learning and life.

### 3.2. Factor II: ACP Does Not Help with Balancing Theory and Clinical Field Practice (Critical View of ACP)

Sixteen (30.8%) of the 52 participants were classified into the factor II group, of which 13 (81.2%) were females and six (37.5%) were 22 years old (Table 3). Compared to the factor I and III groups, the factor II group agreed most (positive perspective) with statements 36 (tuition refund required) and 41 (worried about nursing job well) with scores of +4. Factor II group disagreed most (negative perspective) with statements 26 (ACP itself is a problem) and 32 (less mental stress) with scores of −4 (Table 1). Table 4 shows examples of the post-sorting narratives on factor II, explaining why the participants sorted the statements from +4 to −4 on the grid. The factor II responses reflected the view that ACP was stressful for learning and frustrating for professional competency.

### 3.3. Factor III: RCP Cannot Be Replaced by ACP (Negative View of ACP)

Of the total 52 participants, 18 (34.6%) were classified into the factor III group, in which 13 (72.2%) were female and 11 (61.1%) were under the age of 21 (Table 3). Compared to the factor I and II groups, the factor III group agreed most (positive perspective) with statements 3 (no opportunity to put theory into clinical practice) and 17 (RCP has a lot more to learn than ACP) with scores of +4. Factor III group disagreed most (negative perspective) with statements 10 (not enough learning tools for ACP) and 14 (online nursing skills were very helpful) with scores of −4 (Table 1). Table 4 displays examples of the post-sorting narratives on factor III, explaining why the participants agreed or disagreed with the statements sorted from +4 to −4 in the grid. In this regard, the factor III responses reflected the view that RCP could not be replaced by ACP as RCP provides the opportunity to directly observe and care for patients.

## 4. Discussion

### 4.1. Factor I: ACP Helps with Balancing Learning and Life (Favorable View of ACP)

The most remarkable feature of the factor I group was that they enjoyed and maintained harmony between ACP and personal life. Those in the factor I group believed that ACP enabled self-directed learning as the learning time was from morning to afternoon. They also felt that it was an economically beneficial learning method that allowed for regular part-time work after school hours. They further believed that ACP is a convenient method of learning because it is possible to immediately seek useful information related to learning using diverse online platforms during ACP in real time. These findings are consistent with those of a study conducted by Park and Seo [22] in South Korea; they reported that 20 third- and fourth-year nursing students who participated in focus group interviews expressed experiences of adapting and growing in a new learning environment. These students positively reported that their confidence in nursing skills increased through the virtual simulation learning included in ACP.

In particular, the students reported that they had to “find their own way” to adapt to ACP, which enabled self-regulated learning. Aldridge and McQuagge [17] reported similar findings, wherein five nursing students who participated in an interview stated that “finding my own way” and “developing my own learning style” could be helpful for self-directed learning. According to a study by Kang [11] who reported the ACP experiences of 12 fourth-year nursing students, ACP applied with simulation had a positive effect on learning immersion, satisfaction, and self-confidence. A study conducted by Kunaviktikul et al. [30] supported the finding that ACP was a time-saving method that blended with the personal work–study balanced lives of students and provided flexible extra rest. Kunaviktikul et al. [30] also questioned whether nursing students would be able to acquire academic integrity, practical nursing skills, clinical competencies, engagement, and partnership through ACP learning. Majrashi et al. [1] emphasized that ACP has the advantage of being able to combine learning and work but that there are limits to directly learning professional nursing skills, which the students were anxious about. Therefore, a robust ACP curriculum strategy tailored to the students is essential to further strengthen self-directed learning and improve the learning method of nursing skills in factor I.

However, it is worth noting that factor I tends to prefer ACP rather than RCP, even during the COVID-19 pandemic (Q 27 and 30 in Table 1; post-sorting narrative in Table 4). There is a concern that this view will continue even after the end of COVID-19, and that it can adversely affect the development of clinical competency among nurses after graduation [4]. There is therefore a need for an ACP learning strategy that can improve clinical competence such as nursing skills, decision making, and problem-solving ability that can be achieved through RCP [5,7].

### 4.2. Factor II: ACP Does Not Help with Balancing Theory and Clinical Field Practice (Critical View of ACP)

The most distinctive features of experiences with ACP in the participants in this group were concerns about a decrease in nursing professional competency as well as dissatisfaction and stress regarding the increase in the volume and number of assignments. The factor II group believed that ACP itself was unreliable because it was not real clinical practice and that this would adversely affect the clinical careers of nursing students after graduation (post-sorting narrative in Table 4). These findings were concurrent with those of Majrashi et al. [1], whose analysis of 13 journal articles showed that distance learning, including ACP, during the COVID-19 pandemic was insecure and stressful both mentally and physically owing to the excess learning load, tremendous assignments, and insufficient learning content. A study conducted in Cyprus by Sofianidis et al. [31] supported these findings, where the authors reported that the majority of students had difficulties doing large amounts of homework and recommended that the mitigation of homework and assignments could be effective in generating students’ interest in online learning. Dziurka et al. [18] interviewed 20 Polish nursing students and reported that they initially felt safe with ACP learning because there was no fear of the COVID-19 infection, but the students expressed concerns about their professionalism in nursing and uncertainty about their ability to adapt to clinical settings after graduation. Lim [13] reported that fourth-year nursing students who were about to graduate had fears about becoming registered nurses owing to the lack of real RCP experience. Excess amounts of learning and homework in ACP can easily exhaust students and negatively affect satisfaction with their major. A decrease in satisfaction with their major over ACP may put nursing students at risk of falling behind in the short term and have detrimental effects on their career identity and nursing professionalism in the long term [7,8].

Another special feature of the factor II group was the refund of tuition due to ACP. Tuition refund is a controversial issue at many universities in South Korea, and Kim [5] reported that the lack of class management and opaque learning evaluations at universities could be the cause. Nursing students with this mindset may be disappointed in college life, which may lead to lower satisfaction with their majors. More importantly, these problems can increase school dropouts, which can result in social and national losses owing to the shortage of medical professionals. In South Korea, where the student population is declining, tuition refunds are a very sensitive issue linked to the existence of a university. Providing scholarships and purchasing nursing supplies with the tuition saved through ACP could be an alternative for this group.

### 4.3. Factor III: RCP Cannot Be Replaced by ACP (Negative View of ACP)

The most notable feature of the factor III group was that they strongly preferred and believed in the traditional RCP learning method. They also expressed that RCP was an irreplaceable learning method and should be conducted under any circumstance. This finding is aligned with a previous study conducted by Suliman et al. [23], who reported that nursing students’ experiences with ACP were helplessness, burden, and exhaustion and that they preferred RCP over ACP. Terzi et al. [32] emphasized that ACP was inappropriate for fostering nursing talents from a long-term perspective. The 111 nursing students who participated in a study conducted by Park and Cho [16] stated that the greatest advantage of RCP was providing direct nursing care through direct contact with the patients, which could improve nursing knowledge and skills as well as communication skills. Similar results were reported in a study conducted in Spain by Ramos-Morcillo et al. [33], in which 32 nursing students interviewed expressed that they would not be able to gain clinical self-confidence and that it would be difficult to find a job without RCP experience. Thus, in-depth discussions between the state and society, universities, and medical institutions may be necessary for nursing students to experience RCP even in the COVID-19 situation.

In particular, the factor III group believed that RCP was imperative as a learning tool for the well-being of the nation and society as well as the clinical careers of nursing students even if they were to be infected with COVID-19. This result is similar to a study conducted in Belgium by Ulenaers et al. [14]; they analyzed the experiences of 665 nursing students who participated in RCP and reported that more than 58% of the students were concerned with COVID-19 infection owing to the insufficient provision of personal protective equipment during RCP. Within the context of factor III, the education and training of medical staff, including nursing students, is an indispensable task. Moreover, COVID-19 is an ongoing international crisis, and no one knows when or where other infectious diseases may suddenly manifest. Adequate supplies of personal protective equipment, proactive COVID-19 testing, and encouraging vaccination against COVID-19 will be strategies for those in the factor III group. It may also be helpful to assign a psychology consultant to reduce concerns about COVID-19 infection during RCP.

## 5. Strengths and Limitations

A strength of this study is that it contributes objectively to categorizing nursing students’ subjective points of view on ACP during the COVID-19 pandemic and presenting nursing strategies for each factor. Although this study explored the ACP experiences of nursing students during the COVID-19 pandemic, it has several limitations. First, the number of study participants was irrelevant as the Q methodology focuses on exploring the subjective viewpoints of individuals. A larger sample size may actually serve as an obstacle to categorizing the viewpoints of the individual research participants. Nevertheless, the small convenience sample used in this study may be considered as a limitation to generalizing its results to nursing students with different cultural and educational backgrounds. Second, the homes and schools of the nursing students who participated in this study were located in various areas, such as metropolitan areas and the provinces. Therefore, the education and online environments of the study participants would have been very different, which may have influenced the results of the study.

## 6. Conclusions and Implications

From the results of the forced distribution of the 41 statements in the Q table by 52 nursing students, three distinct factors were derived. These three factors can contribute to the understanding of South Korean nursing students’ experiences, attitudes, and perceptions of the ACP learning method during the COVID-19 pandemic. The 41 statements and post-sorting narratives may also help identify the personal views of South Korean nursing students on RCP and ACP for reflection in nursing education in the future. In particular, nursing strategies designed based on the three factors derived from this study are expected to be helpful in planning ACP in the event of an outbreak of an infectious disease such as COVID-19. The nursing strategy presented in this study is also expected to be helpful in developing a customized ACP when RCP is not available due to a lack of clinical practice institutions. The findings of this study have important implications for nursing educators and educational policy makers from different cultural and educational backgrounds as well as nursing supervisors in clinical placements to encourage and support the clinical practicum of nursing students during the COVID-19 pandemic.

## Figures and Tables

**Figure 1 healthcare-11-02452-f001:**
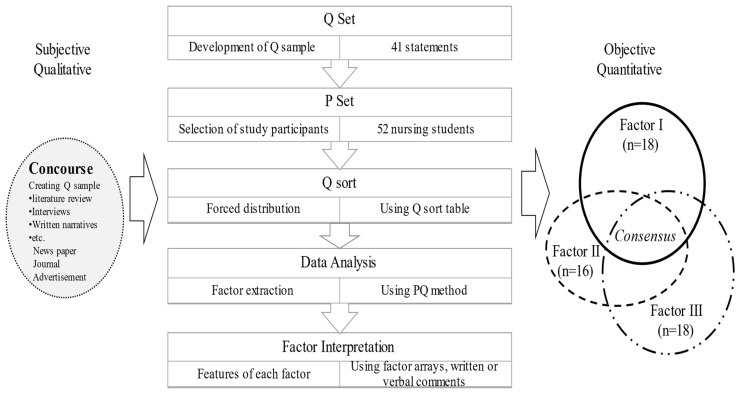
The Study Procedure and Emerging Factors.

**Table 1 healthcare-11-02452-t001:** Factor Arrays (Item by Ranked Position) (*N* = 52).

Q-Statements	I	II	III
	(n = 18)	(n = 16)	(n = 18)
1. There was plenty of time to learn the theoretical basis for the core nursing skills.	**+2**	0	0
2. There were many unexpected situations during the ACP.	−2	**+1**	−2
3. There was no opportunity to put the theory into clinical practice.	+1	**−1**	**+4**
4. I felt like I was learning nursing theory rather than nursing practice.	+2	0	+2
5. I felt more comfortable and at ease that the daily learning contents was set.	**+4**	**−1**	0
6. It was nice to be able to learn independently without having to argue with friends.	+1	−1	−1
7. ACP had a lot of learning quantity to be done, but the learning quality was low.	0	−1	+1
8. The sense of learning achievement for nursing practice decreased.	**−3**	−1	**+1**
9. The amount of homework was so much that it was stressful both mentally and physically.	−1	**+2**	0
10. **There were not enough learning tools needed for ACP, which made the study difficult.**	−3	−2	−4
11. **The professors were not proficient in online media, which made it difficult to learn.**	−2	−3	−2
12. Development of various educational contents and a standardized evaluation for ACP are needed.	+1	+2	+2
13. It is necessary to build an advanced online education environment system.	+1	+1	+1
14. Online Nursing Skills were very helpful.	0	0	**−4**
15. A personal nursing skill kit for home use is required for skill practice.	−1	−1	**+2**
16. It was possible to use of various media that can solve the unknowns immediately during ACP.	+3	0	**−2**
17. Although physically tired, RCP has a lot more to learn than ACP.	**+1**	+2	**+4**
18. There was no vividness that could be experienced in the RCP.	0	0	**+3**
19. **It was disappointing that there was little interaction between professors and students.**	−3	−3	−2
20. There was no opportunity to discuss and share experiences with fellow students.	−2	−3	0
21. **ACP was a learning that could not experience the characteristics of each ward in a hospital.**	+2	+3	+3
22. I was uneasy about whether I was doing it right because I could not directly observe and provide nursing care for patients.	0	+3	+2
23. **It was disappointing that I did not have the opportunity to observe the nursing activities.**	+3	+3	+3
24. **It was disappointing that I could not experience various hospitals.**	+2	+2	+3
25. There was no opportunity to improve critical thinking and therapeutic communication ability.	−2	−2	**+1**
26. The idea of replacing RCP with ACP itself is a problem.	−3	−4	**0**
27. I don’t want to do the ACP again.	**−4**	−3	**−1**
28. ACP was safe because there was no risk of infection with COVID-19.	**+4**	+1	**0**
29. My health deteriorated because I spent a lot of time looking at the computer screen.	−1	0	−2
30. Exceptions should be made so that RCP can be conducted even in the COVID-19.	**−4**	−2	**+1**
31. A balance between quarantine and RCP is necessary. RCP is also learning.	0	+1	+1
32. My body and mind were comfortable, so there was less mental stress.	0	**−4**	−3
33. It was good that ACP did not require a COVID-19 test.	+1	**+1**	**−3**
34. I was able to use my time efficiently.	**+3**	−2	**−1**
35. It was nice to get a part-time job using the remaining time after the ACP.	**+2**	−2	**−3**
36. Tuition fee refund is required for online practice.	0	**+4**	0
37. ACP could save time and money.	+3	+2	+2
38. ACP is economical because there is no practice fee to be paid to the hospital.	−1	0	−1
39. There seems to be a limit to employment due to lack of proper clinical practice.	−2	**+1**	−3
40. Due to the lack of clinical practice experience, the ability to adapt to clinical setting after graduation is likely to decrease.	**−1**	**+3**	−1
41. I have little experience in clinical practice, so I am worried about whether I will be able to do nursing job well after graduation.	**−1**	**+4**	−1

ACP = alternative clinical practice; RCP = routine clinical practice; COVID-19 = coronavirus disease 2019; Bold numbers = Distinguishing statement significant at *p* < 0.001; Bold sentences = three factors’ consensus statement.

**Table 2 healthcare-11-02452-t002:** An Example of a Completed Q sort (No. 31 Participants).

				30					
			35	1	9				
		29	33	31	15	3			
	11	2	38	34	22	25	24		
	10	8	19	13	18	26	41		
32	28	5	36	12	37	7	17	23	Arrangement of 41 Q cards
14	16	6	20	27	39	21	4	40	
−4 (2) 1	−3 (4) 2	−2 (5) 3	−1 (6) 4	0 (7) 5	1 (6) 6	2 (5) 7	3 (4) 8	4 (2) 9	RS (No. of cards) TS
Strongly disagree				Neutral				Strongly agree	

Note. RS = raw score; TS = transformed score; No = number.

**Table 3 healthcare-11-02452-t003:** Demographic Characteristics (*N* = 52).

Characteristics	Categories	Factor I (n = 18) (%)	Factor II (n = 16) (%)	Factor III (n = 18) (%)
Gender	Male	5 (27.8)	3 (18.8)	5 (27.8)
	Female	13 (72.2)	13 (81.2)	13 (72.2)
Age	Below 21	6 (33.3)	5 (31.3)	11 (61.1)
	22	5 (27.8)	6 (37.5)	4 (22.2)
	23	3 (16.7)	2 (12.5)	1 (5.6)
	Over 24	4 (22.2)	3 (18.7)	2 (11.1)
Year	3rd	4 (22.2)	5 (31.3)	6 (33.3)
	4th	14 (77.8)	11 (68.7)	12 (66.7)
Residence	Urban	10 (55.6)	12 (75.0)	7 (38.9)
	Near school	1 (5.5)	1 (6.3)	2 (11.1)
	Rural	7 (38.9)	3 (18.7)	9 (50.0)
Commuting	Home	6 (33.3)	4 (25.0)	4 (22.2)
	Dormitory	10 (55.6)	11 (68.7)	12 (66.7)
	Rented	2 (11.1)	1 (6.3)	2 (11.1)
Economic level	Good	2 (11.1)	1 (6.3)	2 (11.1)
	Average	15 (83.3)	14 (87.4)	16 (88.9)
	Poor	1 (5.6)	1 (6.3)	0 (0)
Thoughts on ACP	Positive	6 (33.3)	5 (31.2)	0 (0)
	Average	12 (66.7)	10 (62.5)	11 (61.1)
	Negative	0 (0)	1 (6.3)	7 (38.9)
Learning tool for ACP	Good	10 (55.6)	8 (50.0)	11 (61.1)
	Average	8 (44.4)	8 (50.0)	7 (38.9)
	Poor	0 (0)	0 (0)	0 (0)
Experience in RCP	Yes	16 (88.9)	14 (87.5)	15 (83.3)
	No	2 (11.1)	2 (12.5)	3 (16.7)
Thoughts on RCP	Positive	6 (33.3)	11 (68.8)	12 (66.7)
	Average	11 (61.1)	3 (18.8)	5 (27.7)
	Negative	1 (5.6)	2 (12.5)	1 (5.6)
School record	Good	2 (11.1)	2 (12.5)	5 (27.7)
	Average	13 (72.2)	12 (75.0)	12 (66.7)
	Poor	3 (16.7)	2 (12.5)	1 (5.6)

ACP = alternative clinical practice; RCP = routine clinical practice.

**Table 4 healthcare-11-02452-t004:** Post-sorting Narratives of Three Factors.

**Factor I: ACP helps with balancing learning and life (favorable view of ACP)** “The amount of ACP per day was set. I was able to lead a planned life, so I was comfortable physically and mentally” (P43) “Learning hours for ACP was fixed, so I was able to get a regular part-time job after the ACP was over. It was a great help to tuition and pocket money”. (P33) “I had a lot of time, so I was able to take the initiative in learning. Best of all, I didn’t have to worry about getting the COVID-19”. (P52)
**Factor II: ACP does not help with balancing theory and clinical field practice (critical view of ACP)** “I couldn’t understand the professor who gave me too much assignment. The ACP was like a class for assignment. I have no idea what I learned because I was in a rush with my assignment. I was like a machine copying homework all day long…. The ACP was more stressful than the RCP. It was a waste of tuition”. (P32) “There was no opportunity for RCP. After graduation, I am really worried about whether I will be able to properly care for patients and whether I will be able to take pride in being a professional. I am afraid of making a mistake with the patient”. (P38) “There were a lot of unexpected situation such as internet disconnection, noise (a pet dog barking, family’s shouts, phone ringing, etc. The class was unprepared for both the professors and students. It was truly a mess, like hustle and bustle…” (P46)
**Factor III: RCP cannot be replaced by ACP (negative view of ACP)** “No matter what the COVID-19 situation, RCP is a must. I don’t understand how ACP replaces RCP. The government and universities are too passive. When a pandemic such as the COVID-19 occurs, it is necessary to create an environment where future nurses can RCP”. (P37) “I could not experience many things that could be learned from RCP, and I could not observe how nurses work. No matter how many advantages of distance learning may be, it is foolish to replace RCP remotely. It is a gamble to teach such a class to a medical person who deals with human life”. (P10) “ACP is not a learning. My dreams and hopes for RCP are gone”. (P26)

ACP = alternative clinical practice; RCP = routine clinical practice; P = participant; COVID-19 = coronavirus disease 2019.

## Data Availability

Due to privacy and ethical restrictions, the data cannot be made public. However, you can check with the author’s email address (rnhaeunho@jwu.ac.kr).

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
