# Peer review of "Alternative Clinical Practice Experiences of Korean Nursing Students during the COVID-19 Pandemic"

_healthcare, 2023, doi:10.3390/healthcare11172452_

Round 1
Reviewer 1 Report
The authors review studies mostly of an empirical type and the methodological proposal called Q, however, there is no theoretical positioning of the same and a hypothesis approach to guide the work they are carrying out. On the other hand, the analysis of the groups created by the authors could be subjected to a cluster analysis to provide statistical significance despite the small size of the sample, which, despite its small size, it is possible to make generalizations, considering the probabilities of error towards the population from which they have been extracted. In the discussion, the first group belonging to the first factor is not described in the same way as the participants belonging to factors two and three. The limitations of the work are limited to the number of participants and there is no indication of the statistical analysis that would allow finding formal differences in the results and a possible generalization with better probabilities. The ethical considerations of the work are respected by the authors since it was submitted to an ethics committee.
Author Response
Response to Reviewer 1 Comments
Journal : Healthcare (ISSN 2227-9032)
Manuscript ID: healthcare-2534626
Type: Article
Title: Alternative clinical practice experiences of Korean nursing students during the COVID-19 pandemic
Section: Nursing
Special Issue: Current Nursing Practice and Education
I sincerely appreciate the reviewer’s attention to detail and the kind comments provided. The detailed response is included below.
Point 1: The authors review studies mostly of an empirical type and the methodological proposal called Q, however, there is no theoretical positioning of the same and a hypothesis approach to guide the work they are carrying out.
Response 1: Thank you for your great comments. Based on your comments, the following modifications have been made. ---> p2
Overview of Q methodology (Q)
The Q methodology was first introduced in 1935 by the psychologist William Stephenson. It is a research method designed to systematically investigate individual subjectivity (thought, attitudes, point of view, perspectives, ideas, etc.) on various issues involving humans [24]. In particular, Q combines qualitative and quantitative methods, and it also has the substantial advantage of being able to quantitatively categorize human internal perspectives [25]. Because of this, Q is widely used as a research method in many areas, including medical service, business, education, advertisement, politics, environment, and economy [26]. However, there is a possible limitation that cannot be ruled out that the steps involved in the Q study can be time-consuming [24].
Point 2: On the other hand, the analysis of the groups created by the authors could be subjected to a cluster analysis to provide statistical significance despite the small size of the sample, which, despite its small size, it is possible to make generalizations, considering the probabilities of error towards the population from which they have been extracted.
Response 2: Q uses a software package specialized in Q-method analysis for factor extraction. Please refer to step 5 for details. However, in deference to your comments, I have added the following sentence: ---> p6
Q method uses software packages such as pc QUANL, PQ method, Ken Q, or PCQ to analyze the data and extract factors.
Point 3: In the discussion, the first group belonging to the first factor is not described in the same way as the participants belonging to factors two and three.
Response 3: Thank you for your great comments. Based on your comments, the following modifications have been made. ---> p9
However, it is worth noting that factor I tends to prefer ACP rather than RCP, even during the COVID-19 pandemic (Q 27 and 30 in Table 1; post-sorting narrative in Table 3). There is a concern that this view will continue even after the end of COVID-19, and that it can adversely affect the development of clinical competency among nurses after graduation [4]. There is therefore a need for an ACP learning strategy that can improve clinical competence such as nursing skills, decision-making, and problem-solving ability that can be achieved through RCP [5,7].
Point 4: The limitations of the work are limited to the number of participants and there is no indication of the statistical analysis that would allow finding formal differences in the results and a possible generalization with better probabilities.
Response 4: Thank you for your great comments. Based on your comments, the following modifications have been made. --->p11
First, the number of study participants was irrelevant as Q methodology focuses on exploring the subjective viewpoints of individuals. A larger sample size may actually serve as an obstacle to categorizing the viewpoints of the individual research participants. Nevertheless, the small convenience sample used in this study may be considered as a limitation to generalizing its results to nursing students with different cultural and educational backgrounds.
** Please see the attachment (revised manuscript by author)

Reviewer 2 Report
Additional reviewer feedback:
1. Font within the footnotes under Tables 1, 2, and 3 are not consistent and numeric item listings are not equivalent and these areas might require some slight revisions.
2. Please address the validity and reliability of Q and PQ measures as applicable.
3. Address the strengths of the study under the discussion section.
4. Study findings report on factors I, II, and III in relation to females. Could the authors provide insight on the significance of study findings for factors I, II, III in relation to males and in comparison to these findings for males versus females (See Table 2 within the manuscript).
Please adhere to and revise any slight English language/grammar errors utilizing the format APA 7Th edition format that is preferable by the journal.
Author Response
Response to Reviewer 2 Comments
Journal : Healthcare (ISSN 2227-9032)
Manuscript ID: healthcare-2534626
Type: Article
Title: Alternative clinical practice experiences of Korean nursing students during the COVID-19 pandemic
Section: Nursing
Special Issue: Current Nursing Practice and Education
I sincerely appreciate the reviewer’s attention to detail and the kind comments provided. The detailed response is included below.
Point 1: Font within the footnotes under Tables 1, 2, and 3 are not consistent and numeric item listings are not equivalent and these areas might require some slight revisions.
Response 1: Thank you for your great comments. All of them have been modified according to your comments (footnotes under Tables 1, 2, 3, and numeric item listings) and presented in the text.
Point 2: Please address the validity and reliability of Q and PQ measures as applicable.
Response 2: Thank you for your great comments. The validity and reliability of the Q methodology was mentioned in the 2.3. Rigor (p6) session. However, I respect your comments. Therefore, the following contents have been added to the text.----> p6
The validity of the Q method is verified through a process involving three stages: content validity, face validity, and Q-sorting validity. The reliability of the Q methodology is verified when there is a Pearson correlation coefficient of .80 or higher through test-retest [26,27].
Point 3: Address the strengths of the study under the discussion section.
Response 3: In deference to your comments, I have added the strengths of this study as follows. ---> p10
- Strengths and limitation
A strength of this study is that it makes the contribution of objectively categorizing nursing students' subjective points of view on ACP during the COVID-19 pandemic and presenting nursing strategies for each factor.
Point 4: Study findings report on factors I, II, and III in relation to females. Could the authors provide insight on the significance of study findings for factors I, II, III in relation to males and in comparison to these findings for males versus females (See Table 2 within the manuscript).
Response 4: Thank you very much for your incisive points. This study investigated the subjective experiences of nursing students, not the differences between male and female nursing students regarding ACP. Therefore, in this study, there are limitations in comparing and analyzing the differences in viewpoints between male and female nursing students. However, since your suggestion is an academically valuable study, it will be reflected in the next study.
** Please see the attachment (revised manuscript by author)

Reviewer 3 Report

Good command of the English language.
Author Response
Response to Reviewer 3 Comments
Journal : Healthcare (ISSN 2227-9032)
Manuscript ID: healthcare-2534626
Type: Article
Title: Alternative clinical practice experiences of Korean nursing students during the COVID-19 pandemic
Section: Nursing
Special Issue: Current Nursing Practice and Education
I sincerely appreciate the reviewer’s attention to detail and the kind comments provided. The detailed response is included below.
Point 1: The purpose and conclusion are not congruent in the abstract. The conclusion in the abstract deals with only the implications of the study and does not reflect the result of the study.
Response 1: Thank you for your great comments. Based on your comments, the following modifications have been made. ---> Abstract section
The findings of this study indicate that developing a curriculum for clinical practice that can enhance the strengths of ACP while compensating for its weaknesses will help promote learning among nursing students.
Point 2: References are needed in the introduction section paragraph 4 pertaining to the COVID-19 pandemic.
Response 2: Is the 4th paragraph you refer to 'Many qualitative and quantitative studies have been conducted globally on the ACP of nursing students during the COVID-19 pandemic'? If so, I modified it to: ---> p2
Many qualitative and quantitative studies have been conducted worldwide to explore ACP among nursing students during the COVID-19 pandemic [12,22,23].
Point 3: The manuscript has some fonts and grammatical errors that need to be addressed.
Response 3: Thank you for your great comments. I have revised both the fonts and grammatical errors you mentioned. A certificate of editing by the professional English editors at HARRISCO is attached in the revised manuscript.
Point 4: Discussion font size is incorrect and varied throughout the discussion.
Response 4: Thank you for your great comments. I have revised both incorrect and varied fonts
Point 5: Limitation section has minor grammatical errors.
Response 5: Thank you for your great comments. Grammatical errors in discuss section have been revised. A certificate of editing by the professional English editors at HARRISCO is attached in the revised manuscript (p14)
** Please see the attachment (revised manuscript by author)

Reviewer 4 Report
Review Report: healthcare-2534626-peer-review-v1
Title: Alternative Clinical Practice Experiences of Korean Nursing Students During the COVID-19 Pandemic
Overall: Overall it is a good article.
Title: The title sounds scientific and well thought out.
Abstract: It is good. Keywords are mentioned.
Introduction: The introduction is aligning with the objectives. The introduction is well written. The Authors have well introduced the concepts of Routine clinical practice (RCP) and alternative clinical practice (ACP). The objectives of the study are mentioned clearly.
Method & Design: The research design used for this study i.e. Q methodology, is appropriate. The all steps followed are correct.
How did you select the sample? How many participants were in focus group interview?
Ethical approval of the research is granted.
The tools used for this study seems to be appropriate. All statistical techniques used are appropriate for this research.
Results: All results have been reported correctly.
Discussion: The discussion section is alright.
Conclusion: The conclusion and implication section needs clarity. It needs clear direction and needs to be further strengthened.
Limitations of study: The researchers had outlined the limitations of the current research.
All the following sections have been satisfactorily answered: Acknowledgments, Author Contributions, Conflicts of Interest, Funding, Data Availability Statement
Note: The font style and size varies. Use standard format and style.
Author Response
Response to Reviewer 4 Comments
Journal : Healthcare (ISSN 2227-9032)
Manuscript ID: healthcare-2534626
Type: Article
Title: Alternative clinical practice experiences of Korean nursing students during the COVID-19 pandemic
Section: Nursing
Special Issue: Current Nursing Practice and Education
I sincerely appreciate the reviewer’s attention to detail and the kind comments provided. The detailed response is included below.
Point 1: How did you select the sample? How many participants were in focus group interview?
Response 1: Thank you for your great comments. Based on your comments, the following modifications have been made. --->p3
Fifty nursing students voluntarily applied for the focus group interview through the J university website, and these participants were then divided into eight groups (six to seven students per group).
Point 2: The conclusion and implication section need clarity. It needs clear direction and needs to be further strengthened.
Response 2: Thank you for your great comments. Based on your comments, the following modifications have been made. ---> p11
In particular, nursing strategies designed based on the three factors derived from this study are expected to be helpful in planning ACP in the event of an outbreak of an infectious disease such as COVID-19. The nursing strategy presented in this study is also expected to be helpful in developing a customized ACP when RCP is not available due to a lack of clinical practice institutions.
Point 3: The font style and size vary. Use standard format and style.
Response 3: Thank you for your great comments. I have revised both incorrect and varied fonts
